# Allocating epidemic response teams and vaccine deliveries by drone in generic network structures, according to expected prevented exposures

**Dean Matter**[ID]*, **Linke Potgieter**

Department of Logistics, Stellenbosch University, Stellenbosch, Western Cape, South Africa

* deanmmatter@gmail.com

**Data Availability Statement:** All relevant data are within the manuscript and its Supporting information files.

## Abstract

The tumultuous inception of an epidemic is usually accompanied by difficulty in determining how to respond best. In developing nations, this can be compounded by logistical challenges, such as vaccine shortages and poor road infrastructure. To provide guidance towards improved epidemic response, various resource allocation models, in conjunction with a network-based SEIRVD epidemic model, are proposed in this article. Further, the feasibility of using drones for vaccine delivery is evaluated, and assorted relevant parameters are discussed. For the sake of generality, these results are presented for multiple network structures, representing interconnected populations—upon which repeated epidemic simulations are performed. The resource allocation models formulated maximise expected prevented exposures on each day of a simulated epidemic, by allocating response teams and vaccine deliveries according to the solutions of two respective integer programming problems—thereby influencing the simulated epidemic through the SEIRVD model. These models, when compared with a range of alternative resource allocation strategies, were found to reduce both the number of cases per epidemic, and the number of vaccines required. Consequently, the recommendation is made that such models be used as decision support tools in epidemic response. In the absence thereof, prioritizing locations for vaccinations according to susceptible population, rather than total population or number of infections, is most effective for the majority of network types. In other results, fixed-wing drones are demonstrated to be a viable delivery method for vaccines in the context of an epidemic, if sufficient drones can be promptly procured; the detrimental effect of intervention delay was discovered to be significant. In addition, the importance of well-documented routine vaccination activities is highlighted, due to the benefits of increased pre-epidemic immunity rates, and targeted vaccination.

**Funding:** The author(s) received no specific funding for this work.

**Competing interests:** The authors have declared that no competing interests exist.

## Introduction

The COVID-19 pandemic proved a stark reminder of the health risks inherent in our increasingly interconnected and fast-paced world. As international travel increases, so does the spread of microorganisms; including those causing life threatening diseases. Despite routine vaccination campaigns making significant inroads into attaining worldwide herd immunity for many infectious diseases, the risk of epidemics remains—particularly for previously unknown diseases. It is essential that governments and humanitarian aid organisations respond to outbreaks with maximal efficiency, to reduce the loss of life. When such responses to epidemics involve the administration of vaccines, response teams and vaccines need to be allocated to affected areas in a sensible manner. This allocation should take into account the impact of proposed vaccinations in each area on the spread of the epidemic, as well as the opportunity cost of not allocating resources to other affected areas.

Another challenge faced in epidemic response is poor road infrastructure, particularly in developing nations. The recent advent of drone technology has promising applications in this regard, since it enables faster vaccine delivery, and deliveries to areas unreachable by land. Drone deliveries can also possibly curb the spread of an epidemic by reducing human interaction. Since approximately only one quarter of roads in Sub-Saharan Africa are paved [1], and many can be impassable in rainy seasons, drones are a particularly attractive prospect for vaccine deliveries to these hard-to-reach areas, which often have low vaccination rates.

There is existing research into resource allocation in the context of epidemics; Hethcote and Waltman [2] used dynamic programming to find an optimal, lowest-cost solution to a deterministic SIR model, with the cost function assumed to include "money, equipment, personnel, supplies, the disruption of health care elsewhere, etc." A general linear programming model for healthcare resource allocation was formulated by Feldstein *et al.*, with resource availability constraints and various objective functions discussed [3]. In a similar vein, Vitoriano *et al.* [4] created a goal programming formulation of the problem of humanitarian aid resource distribution, accounting for objectives including cost and response time. Zaric and Brandeau [5] presented a dynamic investment allocation model with heuristics, with the objective of maximising certain health benefits. In previous related work, Zaric and Brandeau [6] also developed an investment allocation model, in conjunction with a compartmental epidemic model, to maximise "life years gained or the number of new infections averted. Duijzer *et al.* [7] attempted to maximise the same metric; namely, the "total number of people who escape infection," by allocating a constrained vaccine stockpile between multiple regions, according to what they term a "dose-optimal vaccination fraction." This fraction, determined for each region, maximises the total number of averted infections per vaccine dose assigned. Mbah and Gilligan [8] also focused on resource allocation between multiple regions, and proposed candidate allocation strategies for optimality, based upon necessary conditions derived using the Pontryagin maximum principle.

As Duijzer *et al.* noted [7], multiple previous studies [9–12] show that "the optimal allocation [of vaccines] depends on the moment of vaccination," rendering many static resource allocation strategies sub-optimal. In groundbreaking work which accommodates for this, Teytelman and Larson [13] presented a "telescope-to-the-future switching heuristic", where the resource allocation strategy is dynamically adjusted as new information about an epidemic becomes available over time. This heuristic determines the best vaccine allocations in a network of locations, for each day in the future until a predetermined point, maximising the total number of averted infections by switching batches of vaccine doses between regions. Additionally, they discussed a "greedy heuristic," where vaccines are allocated to locations according to the largest marginal benefit arising per vaccine, where the benefit is averted infections.

Coupled with the aforementioned research into epidemic resource allocation, the recent emergence of drone technology has caused a flurry of research into the possibility of drone delivery for medical purposes, although none regarding delivery allocation during an epidemic. Murray and Chu considered the case of drones being launched from delivery vehicles, naming this problem the 'flying sidekick travelling salesman problem (FSTSP)' [14]. Poikonen *et al*. named this same type of delivery network a 'vehicle routing problem with drones (VRPD)' [15]. Murray and Chu also formulated a 'parallel drone scheduling TSP (PDSTSP)', to model an alternative drone delivery scenario; where vehicle deliveries are performed along a TSP route, and supplementary drone deliveries are made directly to demand nodes from a distribution centre [14]. Finally, Scott and Scott modelled a different delivery network, consisting of vehicle deliveries to 'drone nests', and drones used for 'last-mile' delivery, with the objective of minimizing total delivery time [16].

Future work proposed by Mbah and Gilligan in the field of resource allocation during an epidemic includes the "allowance of heterogeneity in the size of sub-populations," and investigation into the robustness of resource allocation strategies to changes in epidemiological parameters [8]. These, along with addressing the World Health Organisation's need for "optimal strategies for reaching hard-to-reach populations [for vaccination]" [17], are among the objectives of this article. The allocation of vaccination teams and vaccine deliveries, to sub-populations in multiple generic population distribution structures, is discussed. An expected prevented exposures (EPE) strategy is proposed for the allocation of each of these resources, and is compared with certain other allocation strategies commonly considered in literature. The feasibility of using drones for vaccine delivery is evaluated, and a range of parameters influencing the spread of an epidemic are discussed.

## Methodology

A network-based SEIRVD epidemic model is formulated, to simulate the spread of an epidemic through a network of interconnected populations. This model consists of distinct systems of difference equations, with one for each population. On each daily time step, a small proportion of each population migrates to other populations in the network, to simulate the spatial spread of the epidemic. The epidemic intervention is incorporated into this simulation through daily vaccinations by response teams, for each population in the network. Various resource allocation strategies are then proposed, with the objective of ensuring that these vaccinations are performed at locations such that the greatest overall health benefit is achieved. Among these are the two EPE strategies, for both team and vaccine delivery allocation, which are expounded in this section.

For the purpose of demonstration, measles epidemics are simulated, although the model is general and can easily be applied to other similar diseases. Fixed-wing drones similar to those used by Zipline are assumed to be used for vaccine deliveries, since these drones can usually fly considerably further than quadcopters before requiring a recharge. For simplicity, deliveries are assumed to only originate from a single distribution centre (DC). The temperature at which the vaccines are transported and stored is not considered, and the entire network of populations is assumed to be closed, with no migration into or out of the network. Other intervention strategies such as travel restrictions, quarantine orders, and educational campaigns to increase turnout for vaccination, can also reduce the spread of an epidemic, but are outside of the scope of this article. The vaccination intervention is assumed to be the only one occurring, with no other healthcare activities considered.

## Epidemic model

The spread of the epidemic through the network of populations is governed by a set of difference equations for each population, as aforementioned. Let $N_{i,t}$ denote the population size at location $i \in \{1, \ldots, n\}$ on day $t \in \{1, \ldots, T\}$. Similarly, let $S_{i,t}$, $E_{i,t}$, $I_{i,t}$, $R_{i,t}$, $V_{i,t}$, and $D_{i,t}$, denote the number of susceptible, exposed, infectious, recovered, vaccinated and deceased individuals in the population of location $i \in \{1, \ldots, n\}$ on day $t \in \{1, \ldots, T\}$, respectively. Then, for each location $i \in \{1, \ldots, n\}$ and day $t \in \{1, \ldots, T\}$, the progression of the epidemic is modelled by

$$S_{i,t+1} = S_{i,t} - \frac{\beta S_{i,t} I_{i,t}}{N_{i,t}} - v_{i,t} \theta_{i,t}^S \lambda_S + \sum_{j=1, j \neq i}^{n} (m_{j,i} S_{j,t} - m_{i,j} S_{i,t}), \tag{1}$$

$$E_{i,t+1} = E_{i,t} + \frac{\beta S_{i,t} I_{i,t}}{N_{i,t}} - \sigma E_{i,t} - v_{i,t} \theta_{i,t}^{E_{72}} \lambda_E + \sum_{j=1, j \neq i}^{n} (m_{j,i} E_{j,t} - m_{i,j} E_{i,t}), \tag{2}$$

$$I_{i,t+1} = I_{i,t} + \sigma E_{i,t} - \gamma I_{i,t} - \mu I_{i,t} + \sum_{j=1, j \neq i}^{n} (m_{j,i} I_{j,t} - m_{i,j} I_{i,t}), \tag{3}$$

$$R_{i,t+1} = R_{i,t} + \gamma I_{i,t} - v_{i,t} \theta_{i,t}^R + \sum_{j=1, j \neq i}^{n} (m_{j,i} R_{j,t} - m_{i,j} R_{i,t}), \tag{4}$$

$$V_{i,t+1} = V_{i,t} + v_{i,t} (\theta_{i,t}^S \lambda_S + \theta_{i,t}^{E_{72}} \lambda_E + \theta_{i,t}^R) + \sum_{j=1, j \neq i}^{n} (m_{j,i} V_{j,t} - m_{i,j} V_{i,t}), \tag{5}$$

$$D_{i,t+1} = D_{i,t} + \mu I_{i,t}. \tag{6}$$

In Eqs (1)–(6), $\beta$ denotes the transmission rate of the disease, which is defined as the basic reproductive number $R_0$, divided by the number of days of infection before recovery. The reciprocal of this duration of infection is used as the daily proportion of infectious individuals who recover, $\gamma$. The parameter $\sigma$ denotes the proportion of exposed individuals becoming infectious on any given day—which, similarly, is the reciprocal of the duration of the latent period between initial exposure and the onset of symptoms (and contagiousness). Finally, the daily death rate (the proportion of infectious people who die each day) is denoted by $\mu$, and is calculated as the case fatality rate, divided by the duration of infection. The values of these parameters are estimated such that the properties of the disease under consideration are accurately replicated in the model.

The movement of individuals between populations is modelled in Eqs (1)–(6) using a migration proportion, $m_{i,j}$, to denote the proportion of the population at location $i$ moving to location $j$ daily. This proportion is defined for each pair of locations $i, j \in \{1, \ldots, n\}$, and is assumed to be constant for the duration of each simulation. Each value of $m_{i,j}$ is calculated using the gravity model of migration presented by Kraemer *et. al* [18]—which considers movement between locations to be directly proportional to the sizes of their populations, $N_i$ and $N_j$, and inversely proportional to the distance between them, $\Delta_{i,j}$. Then, where $T_{i,j}$ denotes the

number of people moving from population $i$ to $j$ each day,

$$T_{i,j} = k_m \frac{\sqrt{N_i N_j}}{\Delta_{i,j}^2}.$$

The parameter $k_m$ is a multiplier which can be adjusted to model the effect of increased or reduced migration on the epidemic. Given the above formulation, the proportion of population $i$ migrating to population $j$ daily, is calculated as

$$m_{i,j} = \frac{T_{i,j}}{N_i}.$$

Importantly, note that the above formulation is most accurate for geographic areas no larger than a few hundred kilometres in diameter. In order to better model international migration, more sophisticated migration models can be utilised.

Another important parameter in Eqs (1)–(6) is $v_{i,t}$, the number of vaccinations performed at location $i$ on day $t$. This parameter is calculated as the minimum between the number of vaccinations which teams can possibly perform, and vaccine stock available:

$$v_{i,t} = \min\{\psi\,\eta_{i,t}, v_{i,t}\}.$$

In the above, the number of daily vaccinations per team is denoted by $\psi$, and the number of vaccination teams allocated to location $i$ on day $t$ is $\eta_{i,t}$. The number of vaccines in stock is denoted by $v_{i,t}$. Note that the latter two parameters, $\eta_{i,t}$ and $v_{i,t}$, are the easiest for decision makers to control, through resource allocation strategies for teams and vaccine deliveries, respectively. Turnout for vaccination is not considered to be a limiting factor on the number of vaccinations, and it is assumed that only asymptomatic members of the population are vaccinated, with infected and deceased individuals excluded. More specifically, only individuals in the S, E, and R categories are given vaccinations.

Untargeted vaccination, which is the practice of performing vaccinations while disregarding potential previous vaccinations received, is sometimes used in outbreak response where there is a high risk of the epidemic spreading [19]. As a result, it is used by default in this article, except where otherwise stated. In Eqs (1)–(6), this is modelled by initially placing individuals who have been previously vaccinated into the R category; thereby making them a part of the subpopulation to be vaccinated. On the other hand, if targeted vaccination is employed, these individuals are initially considered part of the V category, and remain there for the duration of the simulation, excluded from the subpopulation to be vaccinated.

The final undiscussed parameters used in Eqs (1)–(6) are those relating to vaccine efficacy. The susceptible proportion of the remaining unvaccinated (and asymptomatic) population is calculated as

$$\theta_{i,t}^S = \frac{S_{i,t}}{S_{i,t} + E_{i,t} + R_{i,t}}.$$

This is essentially the probability that any individual arriving to be vaccinated is susceptible, and neither exposed nor recovered. Likewise, the ratio of recently-exposed individuals (exposed within 72 hours prior to vaccination) to the remaining unvaccinated and

asymptomatic population, at location $i$ on day $t$, is:

$$\theta_{i,t}^{E72} = \frac{\sum_{k=t-3}^{t} \frac{\beta S_{i,k} I_{i,k}}{N_{i,k}}}{S_{i,t} + E_{i,t} + R_{i,t}}.$$

These recently-exposed individuals can be successfully immunized against measles, with the success rate thereof denoted by $\lambda_E$. The success rate of vaccinations given to susceptible individuals is denoted by $\lambda_S$. As stated above, each day's vaccines, aside from those given to susceptible and exposed individuals, are administered to individuals in the R category. Thus, similarly to the previous two ratios, the proportion of vaccines given to recovered individuals is:

$$\theta_{i,t}^{R} = \frac{R_{i,t}}{S_{i,t} + E_{i,t} + R_{i,t}}.$$

Naturally, all vaccinations given to already-immune individuals can be considered successful, so the efficacy of vaccinations given to these recovered individuals is assumed to be 100%, and all of them are moved from the R category to the V category.

## EPE strategy for team allocation

The number of response teams at each location, $\eta_{i,t}$, is one of the primary decision variables with a direct impact on $\nu_{i,t}$, in the epidemic model formulated. In this section, a resource allocation model is presented, for response team allocation between multiple subpopulations within a network. An integer programming problem is solved on each simulated day, to maximise expected prevented exposures (EPE) for the following day, by assigning teams to locations according to the highest marginal benefit arising from each assignment. This allows the allocation of teams to be dynamically adjusted as the epidemic progresses, ensuring that the allocation remains effective. This dynamic approach, and the objective of minimising total cases by considering the marginal benefit of allocations, is similar to that of Teytelman and Larson's telescope-to-the-future and greedy heuristics for vaccine allocation [13]. However, in this article, response teams and drone deliveries (under the constraint of limited delivery time) are allocated, instead of vaccines (under the constraint of a limited vaccine stockpile and uncertainty about future stock). Allocating deliveries, instead of simply vaccine stock, takes time into account—which ensures that delivery time is most effectively used in the critical first few days and weeks of an epidemic intervention. Another difference of note is that these allocations are performed in multiple generic network structures with heterogeneous population distributions. Their approaches can be seen as higher-level vaccine allocation strategies, perhaps to regions within a country, whereas the strategies in this article are more focused towards the actual delivery of vaccines to allocated response teams at points of dispensing.

Assuming that there are $\mathcal{N}$ vaccination teams in the network, a number $\eta_{i,t}$ of teams needs to be allocated to each location $i \in \{1, \ldots, n\}$ on day $t$, such that $\sum_{i=1}^{n} \eta_{i,t} = \mathcal{N}$, $\forall t \in \{1, \ldots, \mathcal{T}\}$. Recall that $\psi$ denotes the number of possible vaccinations per team, per day. In order to remove the dependency between team allocation and vaccine allocation, the number of vaccines in stock at each location is assumed to be unlimited in the calculation of EPE. With this assumption, and if $\eta_{i,t}$ teams are assigned to location $i$, the number of vaccinations possible at location $i$ is:

$$\nu_{\eta_{i,t},i,t} = \psi \eta_{i,t}.$$

Given this assignment, the number of susceptible individuals remaining, after vaccination at location $i$ on day $t$, is

$$S'_{i,t} = S_{i,t} - v_{\eta_{i,t},i,t} \theta^S_{i,t} \lambda_S.$$

Then, with new exposures, progression into the Infectious category, and vaccines administered taken into account, the expected exposures at location $i$ on day $t + 1$ is

$$E'_{\eta_{i,t},i,t+1} = E_{i,t} + \frac{\beta S'_{i,t} I_{i,t}}{N_{i,t}} - \sigma E_{i,t} - v_{\eta_{i,t},i,t} \theta^{E_{72}}_{i,t} \lambda_E.$$

Now, for each location $i \in \{1, \ldots, n\}$, and each number of teams $\eta_{i,t} \in \{1, \ldots \mathcal{N}\}$ that could possibly be assigned to location $i$, the EPE resulting from that team assignment is calculated as:

$$E^*_{\eta_{i,t},i,t+1} = E'_{0,i,t+1} - E'_{\eta_{i,t},i,t+1}.$$

Given the above values of EPE, for each possible team assignment at each location $i \in \{1, \ldots, n\}$, a knapsack problem is formulated, with the objective of maximising total EPE in the network, by allocating teams for day $t + 1$. The problem is to

$$\text{maximise} \qquad \sum_{i=1}^{n} E^*_{\eta_{i,t},i,t+1} \tag{7}$$

$$\text{s.t.} \qquad \sum_{i=1}^{n} \eta_{i,t} \leq \mathcal{N}, \tag{8}$$

$$\eta_{1,t}, \eta_{2,t}, \ldots, \eta_{n,t} \text{ integer.} \tag{9}$$

On each simulated day $t \in \{1, \ldots, \mathcal{T}\}$, this knapsack problem is solved by iteratively assigning teams to the locations with the highest increase in EPE resulting from each newly-added team. This way, the greatest ratio of payoff (EPE) to cost (using one of the available teams) is repeatedly selected, until there are no teams remaining. If Eqs (7)–(9) is reformulated as a 0/1 knapsack problem, as any integer knapsack problem can be [20, p. 524], this greedy solution procedure is akin to the branch and bound method proposed by Kolesar [21], which produces an optimal solution for 0/1 knapsack problems when the constraint is binding [22]. Therefore, the proposed solution procedure results in an optimal team allocation each day, in terms of maximised EPE for the whole network.

The exact formulation of the solution procedure follows. Initially, $\eta_{i,t} = 0, \forall i \in \{1, \ldots, n\}$. Then, the first team is allocated to the location

$$i' = \max_i \left\{ \frac{E^*_{(\eta_{i,t} + 1),i,t+1}}{(\eta_{i,t} + 1)}, i \in \{1, \ldots, n\} \right\}.$$

Once this team is allocated, $\eta_{i',t}$ is incremented. Each subsequent team is allocated in the same manner, until there are no unassigned teams remaining.

## EPE strategy for vaccine delivery allocation

Once response teams have been allocated to locations in the network affected by the epidemic, it is imperative to provide these teams with an immediate and uninterrupted supply of vaccines. The use of drones to do so can be significantly faster than vehicles, especially for remote

areas with poor or non-existent transport infrastructure. The problem of allocating vaccine resources to locations in the affected area is similar to that of team assignment, except that, with a limited number of drones, the constraint is the amount of delivery time available each day, rather than the number of teams. Given this, a similar formulation of EPE is presented; in this case, EPE is calculated as the number of expected prevented exposures resulting from each drone delivery to each location. Then, another knapsack problem is presented, to maximise total EPE in the network by allocating drone deliveries under the constraint of limited daily delivery time. These deliveries increase $v_{i,t}$, the quantity of vaccine stock available at each location delivered to. This, in turn, may increase $v_{i,t}$, the number of vaccinations performed at that location, and can thereby curb the spread of the epidemic.

Vaccine deliveries to locations are dependent on the number of teams allocated there, $\eta_{i,t}$, the quantity of vaccines already in stock, $v_{i,t}$, and the number of days for which those vaccines will remain potent outside of the cold chain before expiry—which is 3 days in the case of measles vaccines [23]. The exact number of vaccines to be used for the following 3 days at each location can naturally not be foreknown, but is estimated according to the number of unvaccinated people remaining in that location, and the vaccination capacity per response team, $\psi$. More formally, the (non-negative) number of vaccines still required—that is, likely to be used but not yet in stock—on day $t$ at location $i$, is

$$v'_{i,t} = \max\{\min\{S_{i,t} + E_{i,t} + R_{i,t},\ 3\psi\eta_{i,t}\} - v_{i,t},\ 0\}.$$

For simplicity, let $x_i$ denote the number of vaccines delivered to location $i$ on the day considered, $t$. Given this quantity, the number of vaccinations possible on day $t$ is the minimum between the assigned teams' total vaccination capacity, and vaccine stock:

$$v_{x_i,i,t} = \min\{\psi\,\eta_{i,t},\ v_{i,t} + x_i\}.$$

The number of expected exposures is calculated using $S'_{i,t}$, defined as the number of remaining susceptible individuals after these $v_{x_i}$, $i$, $t$ vaccinations have been performed on day $t$, but before migration and epidemic progression for day $t + 1$ are accounted for:

$$S'_{i,t} = S_{i,t} - v_{x_i,i,t}\,\theta^S_{i,t}\,\lambda_S.$$

Taking $S'_{i,t}$ into account, the expected exposures on day $t + 1$ at location $i$, after the $x_i$ vaccines were delivered on day $t$, is

$$E'_{x_i,i,t+1} = E_{i,t} + \frac{\beta S'_{i,t} I_{i,t}}{N_{i,t}} - \sigma E_{i,t} - v_{x_i,i,t}\,\theta^{E_{72}}_{i,t}\,\lambda_E.$$

Given that $E'_{x_i,i,t+1}$ exposures are expected after the vaccine deliveries (and resulting vaccinations) have happened on day $t$, the expected prevented exposures (EPE) caused by the delivery of $x_i$ vaccines is

$$E^*_{x_i,i,t+1} = E'_{0,i,t+1} - E'_{x_i,i,t+1}, \quad \forall i \in \{1,\dots,n\}.$$

Let $d_i$ denote the number of drone deliveries to location $i$ on the day considered, and let $c$ denote the number of vaccines which can be delivered per flight. Then, the total number of vaccines delivered to location $i$ on the day considered is $x_i = cd_i$. Further, let $\delta_i$ denote the return flight time in minutes, for a drone flight from the DC to location $i$ and back. Across all locations, the total number of daily deliveries is constrained by the total available minutes of delivery time per day, $\mathcal{H}$. Subject to this constraint, the allocation of deliveries to locations, which maximises the total number of expected prevented exposures in the network, may then

be found by solving

$$\text{maximise} \quad \sum_{i=1}^{n} E^{*}_{x_i, i, t+1}, \tag{10}$$

$$\text{s.t.} \quad \sum_{i=1}^{n} d_i \delta_i \leq \mathcal{H}, \tag{11}$$

$$x_i \leq v'_{i,t} + c - 1, \quad \forall i \in \{1, \dots, n\} \setminus \{i^*\}, \tag{12}$$

$$d_1, d_2, \dots, d_n \text{ integer.} \tag{13}$$

The daily time available for drone deliveries, $\mathcal{H}$, is the primary constraint on this problem. As aforementioned, this is in contrast to Teytelman and Larson's greedy heuristic and other literature, where vaccine stock is the primary constraint; here instead, the focus is on allocating the limited amount of daily delivery time best. The other constraint, (12), ensures that no drone deliveries will occur to locations for which the amount of vaccines still needed there, $v'_{i,t}$, is already exceeded. Furthermore, since the distribution centre, location $i^*$, is assumed to have unlimited vaccine stock, it is excluded from the delivery network and the problem formulated above.

This integer program can be solved by iteratively assigning deliveries to locations—incrementing one $d_i$ value and adding $c$ to the vaccine stock there, $v_{i,t}$—where the location to be delivered to each time has the highest resultant EPE per minute of delivery time, of all locations. More explicitly, this location is

$$i' = \max_i \left\{ \frac{E^{*}_{c, i, t+1}}{\delta_i}, i \in \{1, \dots, n\} \right\}.$$

Once a delivery is allocated and stock levels are updated accordingly, the EPE value is recalculated for that location (to become the EPE resulting from another delivery), and the next best location is selected to receive the next delivery. As in the team allocation problem's solution procedure, the highest payoff (EPE) per unit of cost (time) is repeatedly selected—although due to the second constraint (12), and the fact that (11) may not always be a binding constraint, the solutions are not necessarily optimal.

## Alternate dynamic resource allocation methods

In order to compare the effectiveness of the two presented EPE resource allocation strategies with other dynamic allocation methods, alternate strategies are presented; for both team allocation, and vaccine delivery allocation. In both cases, the strategies essentially either prioritize total population, susceptible population, or infected population—with varying implementations according to the resources being allocated. Teams are allocated to locations proportionally to the considered population size at each, and vaccines are delivered to locations in an order of priority set by these population sizes. The strategy of proportional allocation according to total population size is commonly known as the "pro-rata" strategy, and since it is generally considered equitable it is often used in practice, and is frequently discussed in literature [7, 8, 13]. The strategies presented here which prioritize susceptible population and infected population are similar to those proposed by Mbah and Gilligan [8]. The alternate methods considered for the allocation of the $\mathcal{N}$ available teams in the network, with non-integer values stochastically rounded up or down, are:

- Team allocation proportional to total population:

$$\eta_{i,t} \simeq \mathcal{N} \times \frac{N_{i,t}}{\sum_{j=1}^{n} N_{j,t}} \, .$$

- Team allocation proportional to susceptible population:

$$\eta_{i,t} \simeq \mathcal{N} \times \frac{S_{i,t}}{\sum_{j=1}^{n} S_{j,t}} \, .$$

- Team allocation proportional to infected population:

$$\eta_{i,t} \simeq \mathcal{N} \times \frac{I_{i,t}}{\sum_{j=1}^{n} I_{j,t}} \, .$$

- Team allocation proportional to each location's ratio of infections to total population:

$$\eta_{i,t} \simeq \mathcal{N} \times \frac{\frac{I_{i,t}}{N_{i,t}}}{\sum_{j=1}^{n} \frac{I_{j,t}}{N_{j,t}}} \, .$$

- An even distribution of teams, where the $\mathcal{N}$ available teams are evenly spread across the $n$ locations using stochastic rounding, with random allocation of leftover teams thereafter:

$$\eta_{i,t} \simeq \frac{\mathcal{N}}{n} \, .$$

As aforementioned, similar strategies are considered for vaccine delivery allocation also. In each of these methods, drone deliveries (of $c$ vaccines per delivery) are repeatedly performed to the selected location $i'$, by incrementing $d_{i'}$, until no more vaccines are required there (i.e., when $v'_{i',t} = 0$). Thereafter, the second-best location is selected to receive deliveries until it no longer requires any more vaccines, and so on. Therefore, the delivery strategies essentially define varying lists of locations in descending order of priority. The considered strategies are:

- Delivery allocation in descending order of total population, where the location $i'$ selected to receive the first delivery is:

$$i' = \max_i \{N_{i,t}, \, i \in \{1, \ldots, n\}\}.$$

- Delivery allocation in descending order of susceptible population, with:

$$i' = \max_i \{S_{i,t}, \, i \in \{1, \ldots, n\}\}.$$

- Delivery allocation in descending order of infected population, with:

$$i' = \max_i\{I_{i,t}, \ i \in \{1, \ldots, n\}\}.$$

- Delivery allocation in descending order of total population, per minute of delivery time. The division by delivery time is performed to ensure that this method also takes it into account when selecting a location. This division is also performed for the EPE delivery strategy. The location $i'$ selected to receive the first delivery is

$$i' = \max_i\left\{\frac{N_{i,t}}{\delta_i}, \ i \in \{1, \ldots, n\}\right\}.$$

- Delivery allocation in descending order of susceptible population, per minute of delivery time, with:

$$i' = \max_i\left\{\frac{S_{i,t}}{\delta_i}, \ i \in \{1, \ldots, n\}\right\}.$$

- Delivery allocation in descending order of infected population, per minute of delivery time, with:

$$i' = \max_i\left\{\frac{I_{i,t}}{\delta_i}, \ i \in \{1, \ldots, n\}\right\}.$$

## Parameters

The SEIRVD model presented in Eqs (1)–(6) is sufficiently general such that it can validly simulate the progression of any communicable disease comprising the categories S, E, I, R, V, and D. More specifically, such a disease is exclusively transmitted through interpersonal contact, has a latent period prior to contagiousness, and results in either death or permanent immunity upon recovery. The parameters appearing in Eqs (1)–(6), such as the daily transmission rate, death rate, and daily proportion of infectious individuals recovering, govern the progression of the disease. For all results in this article, except where otherwise stated, the values used for these parameters are set to simulate the spread of measles—and are listed in Table 1, along with other default parameter values. In order to simulate another disease, alternate values for these parameters must be estimated from literature, or from data about the progression of epidemics of that disease. The default team allocation strategy is pro-rata allocation proportional to total population, and the default vaccine delivery strategy prioritizes locations in order of descending population size. Untargeted vaccination and the monocentric network type are both used by default, unless otherwise stated.

The epidemic model requires an input dataset containing information about a physical network of locations in a hypothetically affected area. This dataset contains Cartesian x- and y-coordinates and population sizes for each location in the network, as well as the number of (exposed, and infectious) index cases at the start of the simulation. The coordinates of each location are programmatically scaled such that the maximum distance between any two locations equals an adjustable parameter $\mathcal{D}$, the diameter of the network. Euclidean distances

**Table 1. Initial set of parameters used in epidemic and intervention simulations.**

| Parameter | Symbol | Value | Reference |
|---|---|---|---|
| Daily proportion of exposed individuals becoming infectious | $\sigma$ | $\frac{1}{10}$ | [24] |
| Basic reproductive number | $R_0$ | 15 | [25] |
| Daily transmission rate | $\beta$ | $\frac{15}{8}$ | [25, 26] |
| Daily proportion of infectious individuals recovering | $\gamma$ | $\frac{1}{8}$ | [26] |
| Migration multiplier | $k_m$ | 1 | - |
| Daily proportion of infectious individuals dying | $\mu$ | $\frac{0.0329}{8}$ | [26, 27] |
| Vaccine efficacy in susceptible individuals | $\lambda_S$ | 95% | [28] |
| Vaccine efficacy in recently-exposed individuals | $\lambda_E$ | 83% | [29] |
| Number of response teams in network | $\mathcal{N}$ | 15 | - |
| Working minutes per day, for deliveries and vaccinations | $\mathcal{H}$ | 660 | [30] |
| Number of vaccines administered per team per day | $\psi$ | 2000 | [30] |
| Number of vaccines delivered per drone flight | $c$ | 60 | [28, 31] |
| Maximum drone flight range, in kilometres | - | 160 | [31] |
| Days of measles vaccine potency before expiry | - | 3 | [23] |
| Proportion of a location's population to be infected before epidemic declaration | - | 0.9% | - |
| Working days per week, for deliveries and vaccinations | - | 7 | - |
| Delay in days between the declaration of an epidemic and intervention starting | - | 15 | [32] |
| Duration of intervention | - | Unlimited | - |
| Number of delivery drones | - | 5 | - |
| Average drone flight speed, in kilometres per hour | - | 100 | [31] |
| Time taken to launch each drone, in minutes | - | 10 | [33] |
| Scaled diameter of monocentric network, in kilometres | - | 40 | - |
| Scaled diameter of polycentric network, in kilometres | - | 100 | - |
| Scaled diameter of city network, in kilometres | - | 20 | - |
| Scaled diameter of rural network, in kilometres | - | 150 | - |
| Population vaccination rate | - | 66% | - |
| Targeted or untargeted vaccination | - | Untargeted | - |

between each pair of locations are calculated based on these coordinates. These distances are used to select the distribution centre (DC) in the manner described below, and to calculate the drone delivery times, from the DC to each location. The maximum flight range of a single drone is 160km [31]. Therefore, under the assumption that drones only deliver to a single location per flight, the single DC must be placed less than 80km away from all locations, if possible.

The total initial population, $N_{i,0}$, at each location $i$, is partitioned into the categories $S_{i,0}$ and $R_{i,0}$, or $V_{i,0}$, depending on the vaccination rate in each area, and on whether or not targeted vaccination is used. The quantity of vaccine stock $v_{i,0}$, and the number of response teams at each location $i \in \{1, \ldots, n\}$, are initially set to zero.

The input required is sufficiently general such that any geographical network of locations can be used as the affected area in the simulated epidemics. For the purposes of this article, four generic network structures are considered that are comparable to real-world population distribution structures: a dense city network with high, homogeneous population density, a sparse rural network with low, homogeneous population density, a monocentric urban network with decreasing density as distance from the centre increases, and a polycentric urban network with multiple centres from which the density reduces as distance from each centre

increases. The input data for each of these four generic networks are supplied in S1–S4 Tables, and the networks are depicted in S4 Fig.

Drone deliveries in each network are assumed to originate from a single DC, with an unlimited supply of vaccines available. This DC is selected from the locations defined in the network, using a minimax facility location problem. This problem is to choose a location $i^*$, such that the maximum distance from $i^*$ to any other location $j \neq i^*$, is minimized. This facility location method is selected, so that as many locations as possible lie within the aforementioned delivery range of drones originating from the selected location. More formally, the DC is placed at

$$i^* = \min_i \max\{\Delta_{i,j}, \; i,j \in \{1, \ldots, n\}, \; i \neq j\}.$$

Using an average flight speed of 100km/h [31], and a fixed launch time of 10 minutes per flight [33], along with the direct distance $\Delta_{i,j}$ between each pair of locations $i$ and $j$ as the flight distance, the return flight time (in minutes), from the DC to each location $i$ and back, is calculated to be

$$\delta_i = 10 + \frac{2\,\Delta_{i,i^*}}{100} \times 60, \quad \forall i \in \{1, \ldots, n\}.$$

## Results

Prior to the presentation of simulation results, a description of the process of validation undertaken, to establish the reliability of the simulation model, is outlaid. Thereafter, simulation results are presented in three sections; firstly, benchmark results using the base set of parameters are outlaid, followed by a comparison of the effectiveness of the resource allocation strategies presented, and a sensitivity analysis of relevant epidemic-related parameters.

### Validation

The validation for the model outlined in this article is performed on a measles outbreak in Niamey, Niger, which lasted from November 2003 until midway through 2004. All parameters and data used for this validation are derived from those used by Grais *et al.* in their presentation of a simulation model on the same epidemic [34], where possible.

Niamey has an approximately monocentric population distribution, and as such the monocentric network structure defined previously is used. In the actual intervention undertaken, children aged 6-59 months were targeted, as they are the most susceptible to measles. Therefore, the population values specified in the input dataset provided in S5 Table refer only to this age group, for which a vaccination rate of 70% is assumed—a total population of 160 000. Distances between locations are scaled such that the diameter of the network is 20km, and an initial 10 index cases are specified in the centre of the network to start the epidemic. In accordance with the proportion estimated by the World Health Organization [35], and the work done by Grais *et al.* [34], only 50% of actual simulated cases are considered 'detected' in the simulations, and are reported on in the validation results.

The simulation of this Niamey outbreak is set to run for 40 weeks, with vaccination being performed for 10 days, as was the case in the actual intervention. In the simulation, an unlimited quantity of vaccine stock is available to the 33 response teams distributed throughout the network, and teams are allocated proportionally to the total population in each location; the common pro-rata strategy described previously. In order to match the total number performed in actuality, 256 vaccinations are performed by each team per day, without regard for

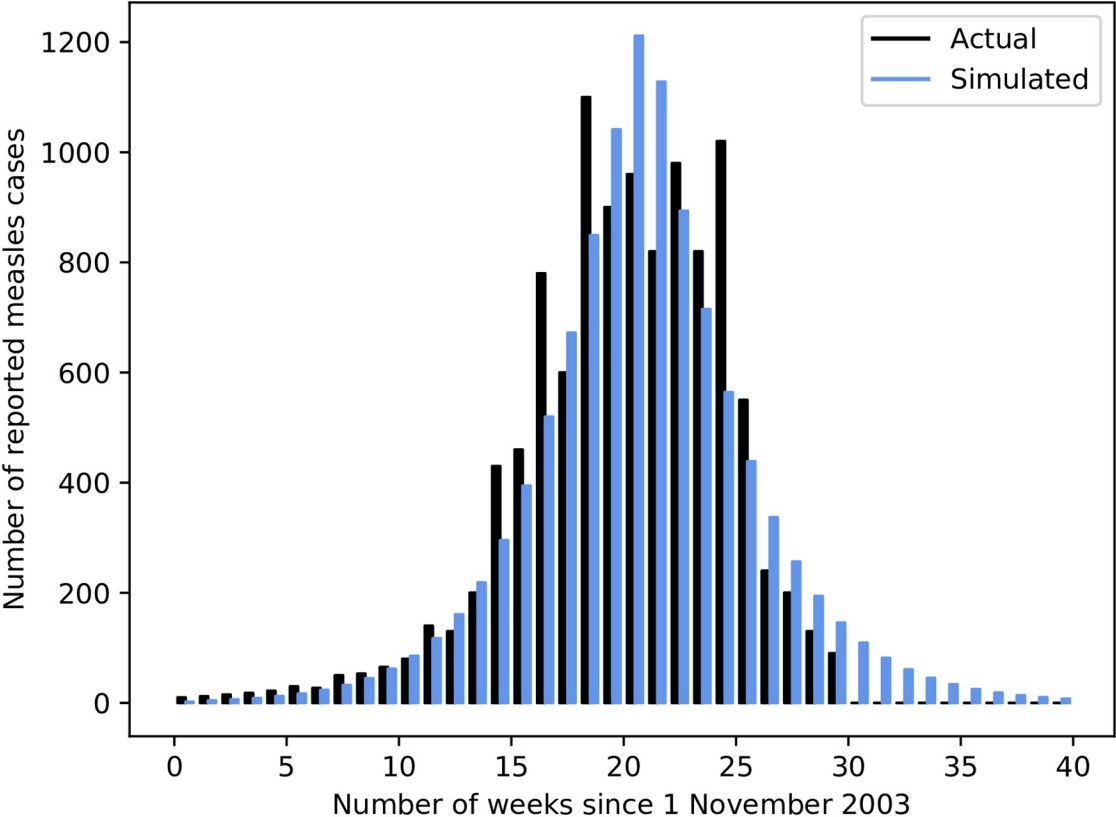

**Fig 1. Comparison between actual and simulated weekly reported cases for Niamey epidemic.**

potential previous vaccinations (i.e. untargeted vaccination is performed). After performing a grid search with repeated simulations with a step size of 0.05, a value of $R_0 = 7.00$ was found to provide the best fit to the reported epidemic curve—and is thus used in the validation simulations. In the same grid search, the ratio of infections to population at the epicentre, when the epidemic is declared, was estimated to be 9 cases in 1000. As a result, this value is used as the epidemic detection threshold for other simulations in this article also.

The best simulated match to the actual epidemic's progression is depicted in Fig 1. Over 100 such simulations, the average number of reported cases per epidemic is 10 857 (with a sample standard deviation of just 0.049)—which is very close to the 10 880 actual recorded cases in Niamey. This was accomplished with exactly 84 480 vaccinations per simulation, near to the 84 563 actually performed. Since the number of teams and daily vaccinations per team are constant values, this number of vaccinations per simulation is also. Given these results, it is clear that the SEIRVD model presented in this article, together with the resource allocation methods used, is able to successfully simulate an actual epidemic.

## Base simulation results

Using the base set of parameters listed in the previous section, simulations were performed using each of the four generic network structures. In order to ensure that results are accurate and that randomness is not an influencing factor, 100 simulations were performed for each network type. Each set of 100 observations, from which the result metrics are calculated, have sufficiently low variance such that the following formula, for the required number of

simulations, equates to less than 100 for all results and metrics:

$$N_{\text{sims}} = \left(\frac{Z_{(1-\alpha/2)} \, S_x}{\epsilon \, x}\right)^2.$$

In this formula, where $x_i$ is the i$^{\text{th}}$ simulation's value for metric $x$, $\bar{x} = \frac{1}{100}\sum_{i=1}^{100} x_i$ is the sample mean of the metric, and $S_x = \frac{1}{100-1}\sum_{i=1}^{100}(x_i - \bar{x})^2$ is the sample standard deviation. The confidence level is $1 - \alpha = 0.95$, and the error level is $\epsilon = 0.05$.

The aggregated results of these simulations for each network are contained in Table 2, and the figures in S5 Fig depict example epidemic progression curves, for a single simulation in each network. It is clear in Table 2 that interventions employing targeted vaccination are more cost efficient (they require fewer vaccinations and drone deliveries), and more effective at reducing the number of cases and subsequent deaths—particularly in less densely populated networks with slower epidemic progression.

## Resource allocation strategy selection

The selected prioritization of certain locations in resource allocation decisions has significant ramifications for the livelihoods of people there. Ethical arguments can be made regarding the fairness of resource allocation to locations with differing socio-economic contexts, but for the purposes of this article, the assumption is made that the target of an epidemic intervention is the reduction of total cases. Therefore, for these resource allocation models and strategies, the minimisation of cases is of first priority, followed by the minimisation of vaccines used. Considering that vaccines are often expensive and in short supply, using the same amount of vaccines more effectively to reduce cases is preferable.

**Comparison of strategy pairs.** Since the team allocation strategies and delivery allocation strategies must be used in conjunction with one another, pairs of strategies are compared; in terms of cases and vaccinations. The figures below contain scatter plots, where each point represents a strategy pair, and is positioned according to the average number of cases and vaccinations occurring in repeated simulations using that strategy pair. The colour of each point represents its team allocation strategy, and the symbol represents its delivery allocation strategy, as represented by the figure legends.

Fig 2 depicts such a scatter plot, for the monocentric network with untargeted vaccination. It is clear that, for this network, the results are somewhat clustered by team allocation strategy,

**Table 2. Simulation results for various network structures with and without vaccination.**

| Network type | Intervention type | Cases | Deaths | Vaccinations | Deliveries |
|---|---|---|---|---|---|
| City | None | 395459 | 12596 | 0 | 0 |
| City | Untargeted | 376483 | 11992 | 1257301 | 17055 |
| City | Targeted | 337983 | 10766 | 600630 | 8356 |
| Rural | None | 60773 | 1936 | 0 | 0 |
| Rural | Untargeted | 34469 | 1098 | 225070 | 3182 |
| Rural | Targeted | 9338 | 298 | 101482 | 1390 |
| Monocentric | None | 202797 | 6459 | 0 | 0 |
| Monocentric | Untargeted | 178297 | 5679 | 724251 | 8782 |
| Monocentric | Targeted | 148102 | 4718 | 462797 | 5798 |
| Polycentric | None | 293785 | 9357 | 0 | 0 |
| Polycentric | Untargeted | 278093 | 8858 | 824228 | 11830 |
| Polycentric | Targeted | 255099 | 8126 | 379448 | 5736 |

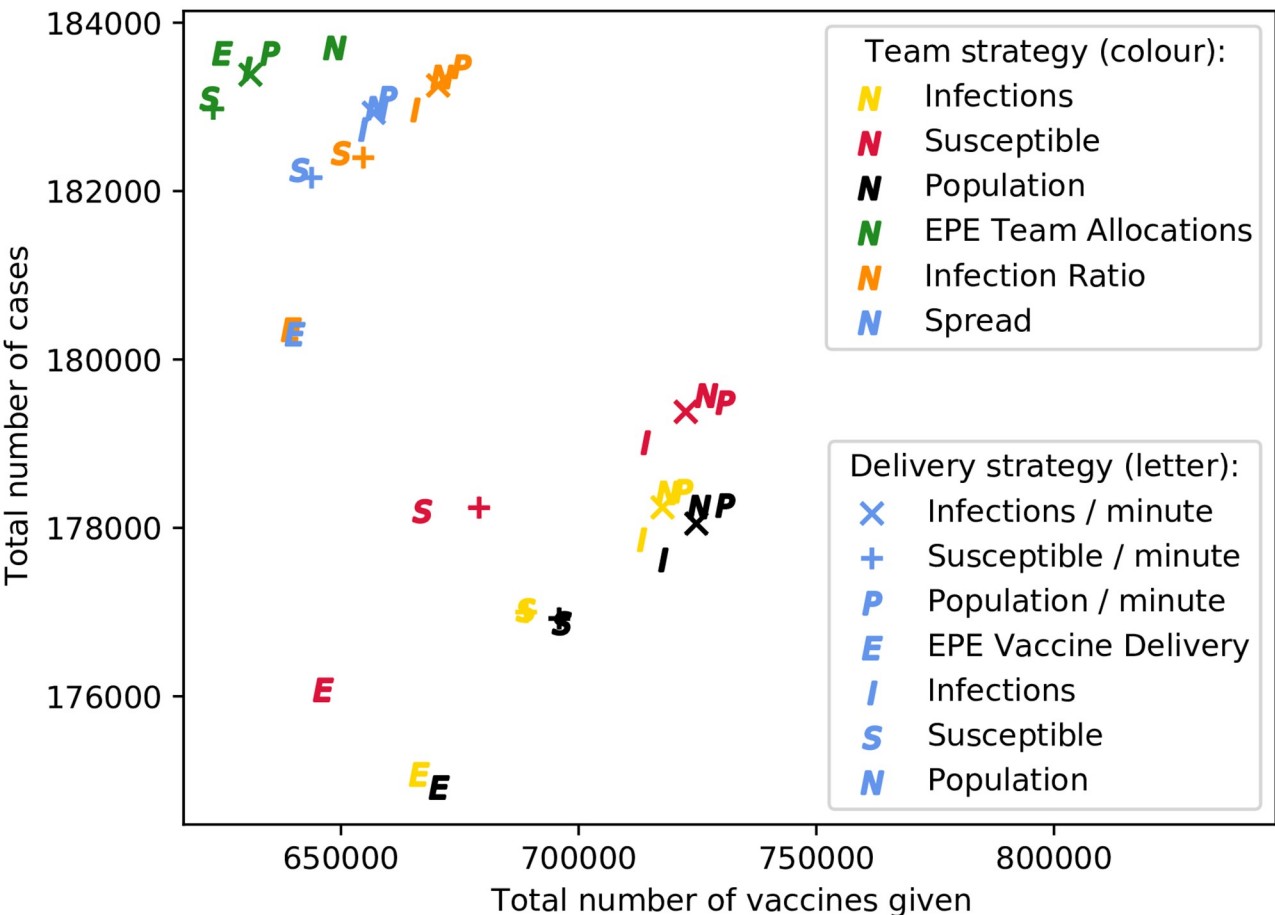

**Fig 2. Comparison of strategy pairs for untargeted vaccination on monocentric network.**

represented by each of the six colours. Evidently, the team allocation strategies resulting in the fewest cases (regardless of vaccine usage) are allocation proportional to total population, susceptible population, and infections. The EPE team allocation strategy performs poorly for this untargeted intervention in the monocentric network, although the number of vaccines given is minimised. In contrast, for the city and rural network types, the EPE team strategy is the best overall, for both targeted and untargeted interventions—highlighting the need for consideration of population distributions when making resource allocation decisions.

Notably, for every team allocation strategy in this network besides EPE, the best delivery allocation strategy is EPE. This is followed by both delivery allocation strategies which prioritise locations by susceptible population. The worst-performing delivery allocation strategies in this network are those prioritising locations according to total population and number of infections. This is aligned with the findings of Mbah and Gilligan [8], Teytelman and Larson [13], and Duijzer *et al.* [7], who all found the pro-rata population strategy to be inferior to other, perhaps less equitable, strategies. A likely hypothesis for the poor performance of the EPE team strategy, considering the superiority of the EPE delivery strategy, is the fact that the primary constraint on vaccinations when using the default parameter set is the amount of available delivery time, rather than the number of teams. Therefore, the selection of an appropriate delivery strategy is more important than that of a team strategy—especially when inefficient untargeted vaccination is employed.

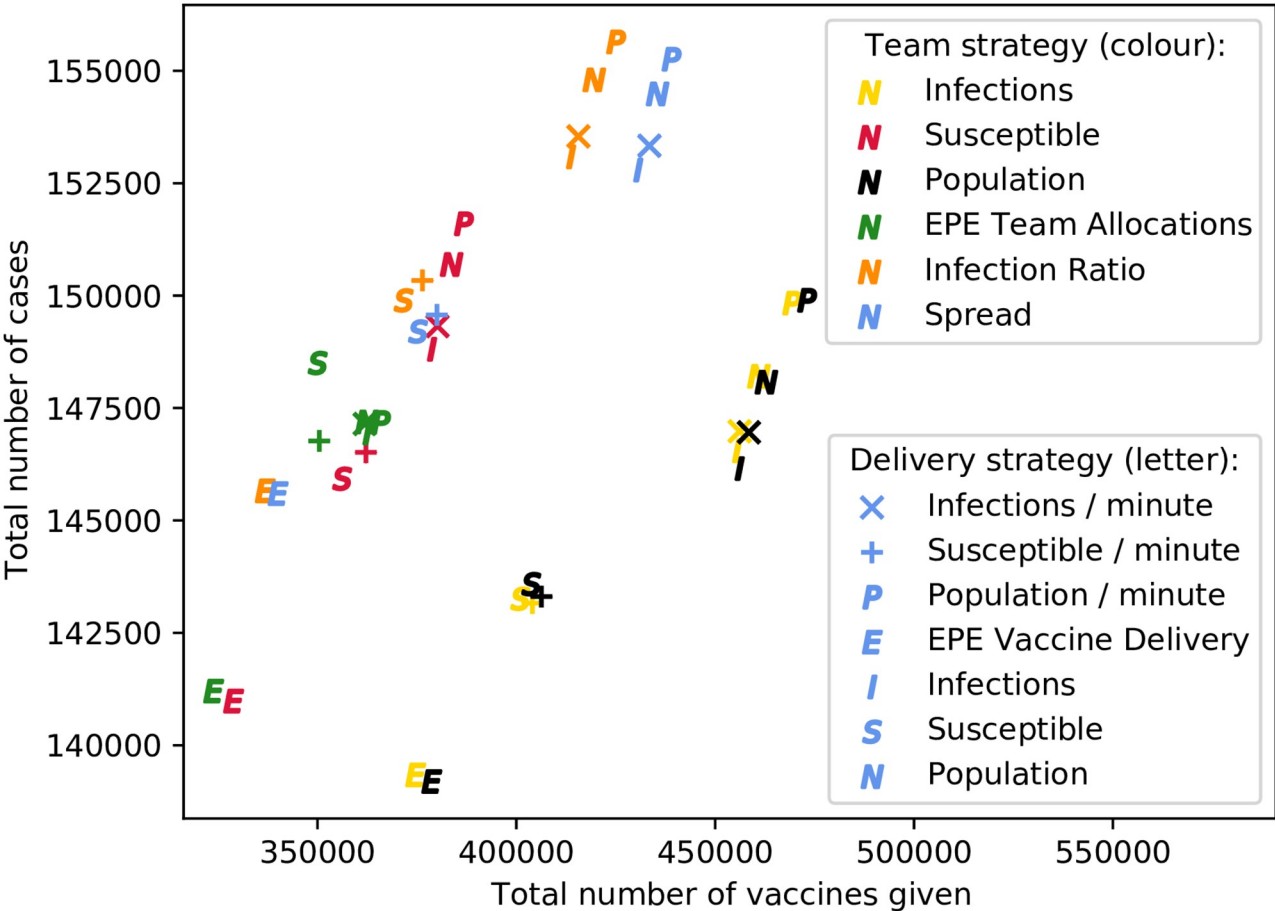

**Fig 3. Comparison of strategy pairs for targeted vaccination on monocentric network.**

Fig 3 contains a scatter plot depicting the results for all strategy pairs, for a targeted intervention on the monocentric network. Once again, the results are largely clustered by team allocation strategy, and the results are similar to the untargeted intervention—with the best strategies being team allocation proportional to population, or infections. If the minimisation of vaccinations is a factor, however, then team allocation according to either EPE, or susceptible population, is superior, since approximately 50 000 fewer vaccines are used, for a marginally higher number of cases. As in the untargeted intervention, prioritising total population and the number of infections for delivery allocation results in an increase in cases, as well as an increase in cost. Note again that the best vaccine delivery strategy, for reducing both cases and vaccinations required, is EPE—for all team allocation strategies, by a significant margin.

Identical scatter plots are also presented for each of the three other network structures considered, in S1–S3 Figs. Each of the other networks have similar results to the monocentric one—the EPE vaccine delivery strategy, for all team allocation strategies, usually results in the fewest number of cases and vaccinations. This superiority of the EPE delivery strategy is more prominent when interventions are targeted. It is clear that prioritising total population for vaccine deliveries to locations is a poor strategy, in comparison with all others presented.

**Comparison of team allocation strategies.** When the base parameter set is used, the primary constraint on vaccinations at locations is vaccine stock availability, due to the limited delivery time available. To prevent this, and the delivery allocation strategies, from influencing

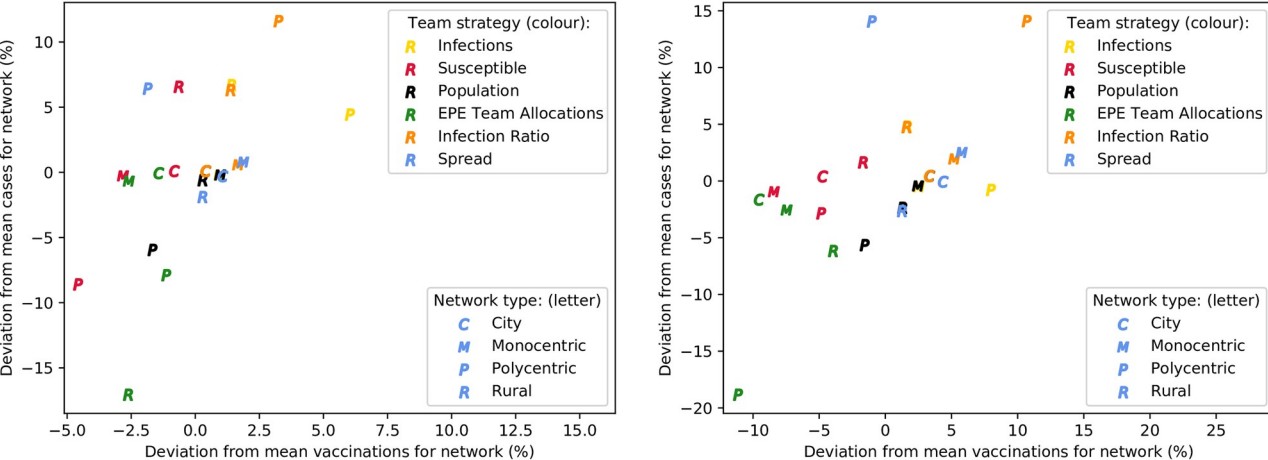

**Fig 4. Comparison of team allocation strategies for differing network types, with unlimited vaccine availability—For untargeted (left), and targeted interventions (right).**

the comparison of team allocation strategies, each location is assumed to have unlimited vaccine stock. Then, repeated simulations are performed for each team strategy and network type, with the mean number of cases and vaccinations recorded for each pair. These are compared with the overall mean values for cases and vaccinations for each network type, such that percentage deviations from the network's means for each metric are calculated. These percentage deviations are used as the coordinates of each point in the scatter plots depicted in Fig 4—representing untargeted and targeted interventions, respectively. These scatter plots thus show an aggregate view of which team strategy performs best for each network type, when vaccine availability is not a constraint.

It is evident that the polycentric and rural networks have the largest difference between team allocation strategies, for both targeted and untargeted interventions. With targeted vaccination, the EPE strategy results in the minimum number of cases for all network types—and this is accomplished with the minimum vaccine usage in all networks besides the monocentric one. Results are similar for the untargeted intervention also, although slightly less prominent. Therefore, particularly where targeted vaccination is used, and sufficient vaccine stock is available, the EPE team allocation strategy is superior to the other strategies considered; minimising both the number of cases, and vaccinations. Aside from the EPE strategy, it is clear that proportional team allocation according to susceptible population is most effective—particularly for reducing vaccine usage, while resulting in a similar, or lower, number of cases than other strategies. Allocating teams proportionally to total population also performs reasonably well, although results in more vaccines being used. Finally, allocating teams proportionally to the number of infections at each location appears to be a costly strategy, and is often less effective in reducing cases. It seems that the effectiveness of certain strategies depends on the network type, and as a result it is useful to consider the spatial population distribution of an area when allocating response teams in an epidemic intervention.

**Comparison of delivery allocation strategies.** A similar comparison to the above is performed for the delivery allocation strategies also. The previously presented base simulation results, using each combination of team and delivery allocation strategies, are used; the results are grouped by delivery strategy and network type, and are aggregated over the team allocation strategies used. Fig 5 depicts similar scatter plots to those previously presented—where, in this case, each point represents the results of repeated simulations using a particular delivery

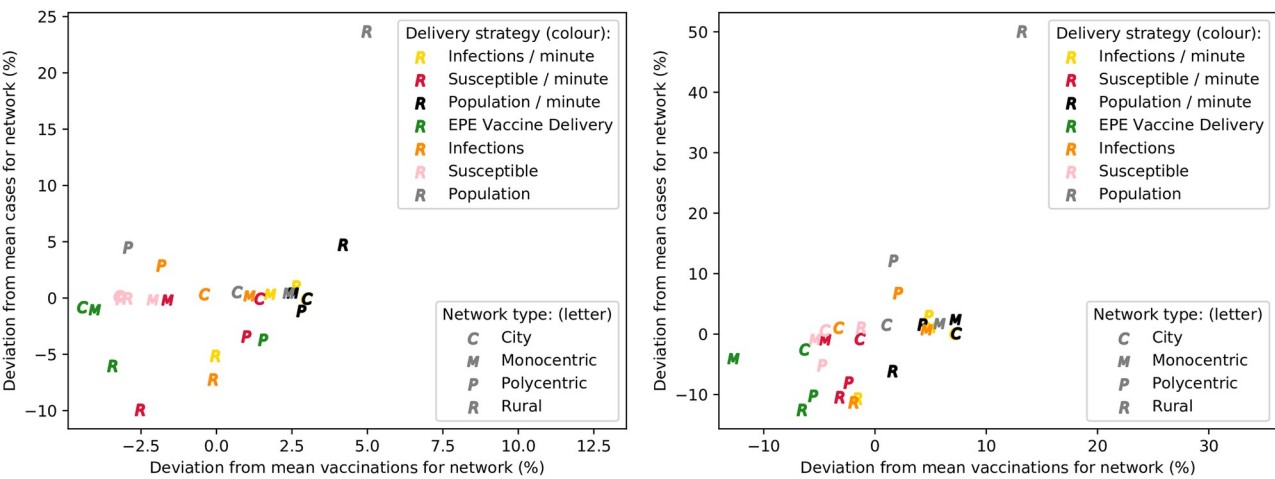

**Fig 5. Comparison of delivery allocation strategies for differing network types, aggregated over team allocation strategies—For untargeted (left) and targeted interventions (right).**

strategy and network type. In these figures, the x- and y- axes represent the deviations of each point's results from the mean number of vaccinations and cases for that network, respectively. This allows the delivery strategies to be more easily compared, in a similar fashion to the team strategies.

As in the team strategy comparison, there is a larger difference between delivery strategies for the polycentric and rural networks than the others. For the rural network, there is a significant outlier; the strategy where vaccine deliveries are performed according to total population. In fact, in all four networks, prioritising either population, or population per minute of delivery time, is an ineffective delivery strategy. In comparison with others, when these strategies are used, vaccine usage is almost always higher, and the number of cases is usually higher also. This is also the case for the two strategies prioritising infected population, albeit to a lesser extent, and with the exception of the rural network type. This is unfortunate since the allocation of vaccines according to population size is seen as an intuitively fair strategy by many decision makers. These results, along with those of Mbah and Gilligan [8], Teytelman and Larson [13], and Duijzer *et al.* [7], indicate that, while apparently equitable and politically acceptable, strategies prioritizing total population are often inferior to others.

In contrast to this, the two delivery strategies prioritising susceptible population usually result in consistently low cases and vaccine usage, for all network types. Selecting locations for deliveries based on susceptible population, per minute of delivery time, is a particularly effective strategy—for many networks, resulting in the minimum number of cases, if the EPE strategy is disregarded. However, if it is possible to implement an EPE vaccine delivery strategy in a targeted epidemic intervention, simulations indicate that it minimises both cost and the number of cases, compared to all other strategies. In untargeted interventions, the EPE strategy is not as clearly superior, but still minimises both cost and cases in three of the four network types considered.

## Sensitivity analysis

A brief sensitivity analysis was performed on certain parameters, to evaluate the effect that changes in their default values have on the simulation results. Each parameter is varied individually, with the remaining parameters constant; for 100 simulations per parameter value. The metrics of most interest that are reported on in this analysis are the number of simulated

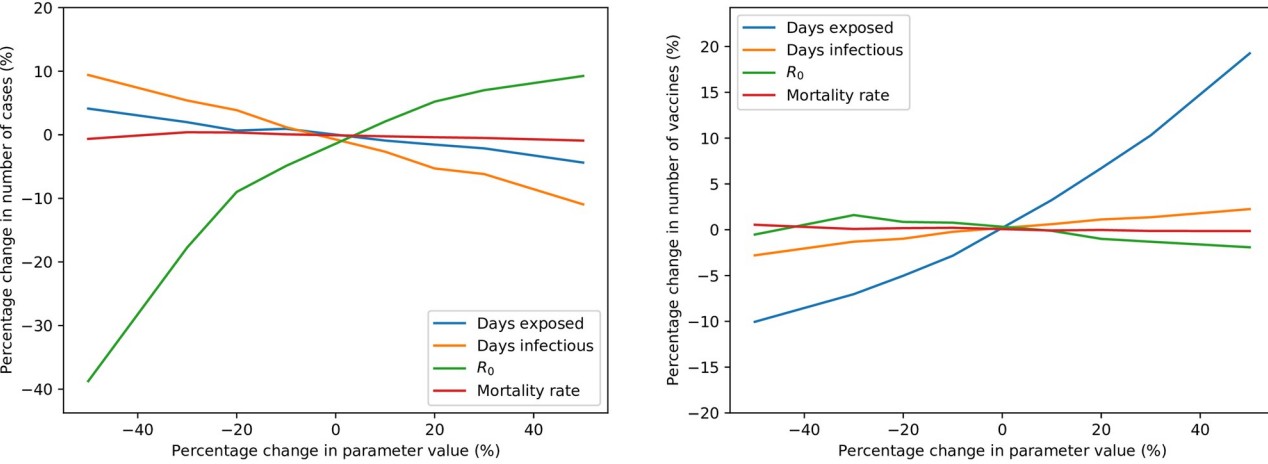

**Fig 6. Sensitivity of average cases (left) and vaccinations (right) to changes in epidemic parameters.**

cases, and the number of vaccines administered—which is directly proportional to the cost of the intervention.

**Epidemic parameters.** Four epidemic-related parameters are considered for sensitivity analysis; the average number of days between initial exposure and becoming infectious, the average number of days being infectious before recovery, the basic reproductive number $R_0$, and the morbidity rate. The effects of changes in these parameters on the average number of cases and vaccines administered are depicted in Fig 6. As both the number of days exposed and infectious increase, the number of cases reduce slightly; likely due to the epidemic progressing more slowly, allowing the intervention to be more timely and effective. As $R_0$ increases, the average number of cases also increases, as expected—albeit increasing at a reducing rate as $R_0$ rises. The final notable result is the apparent near-exponential growth in the number of vaccines administered as the average number of days exposed rises. This is most likely the consequence of vaccines being ineffective on most asymptomatic, exposed individuals; the longer the duration of exposure, the more vaccines are wasted on exposed individuals.

**Intervention parameters.** The second set of parameters investigated are those relating to the intervention itself; the number of working hours for deliveries and vaccinations per day, the number of daily vaccinations per response team, the delay before an epidemic is declared, and the delay between epidemic declaration and the start of the intervention. The sensitivities of both cases and vaccinations to changes in these parameters are plotted in Fig 7.

The modelled effect of an increase in working hours is a proportional increase in daily vaccinations per team, as well as allowing more daily drone deliveries to be possible. Therefore, when either the daily working hours or the daily vaccinations per team is increased, the resultant near-identical slight reduction in cases is expected. The two most notable results for this parameter set are the respective impacts of epidemic detection delay, and intervention delay, on the number of cases. If the epidemic is detected earlier, there is a large reduction in the number of cases. The same applies for intervention delay after epidemic detection; the sooner the intervention, the fewer cases occur on average. Epidemic detection delay is near-impossible for decision-makers in an epidemic intervention to influence, and can only be reduced by more effective routine epidemic monitoring in healthcare systems. On the contrary, while difficult to reduce, the intervention delay is somewhat under the control of decision-makers. The effect of the intervention delay on the number of cases is so pronounced that, for each further day of delay (in the monocentric network), the number of cases in the epidemic increases by

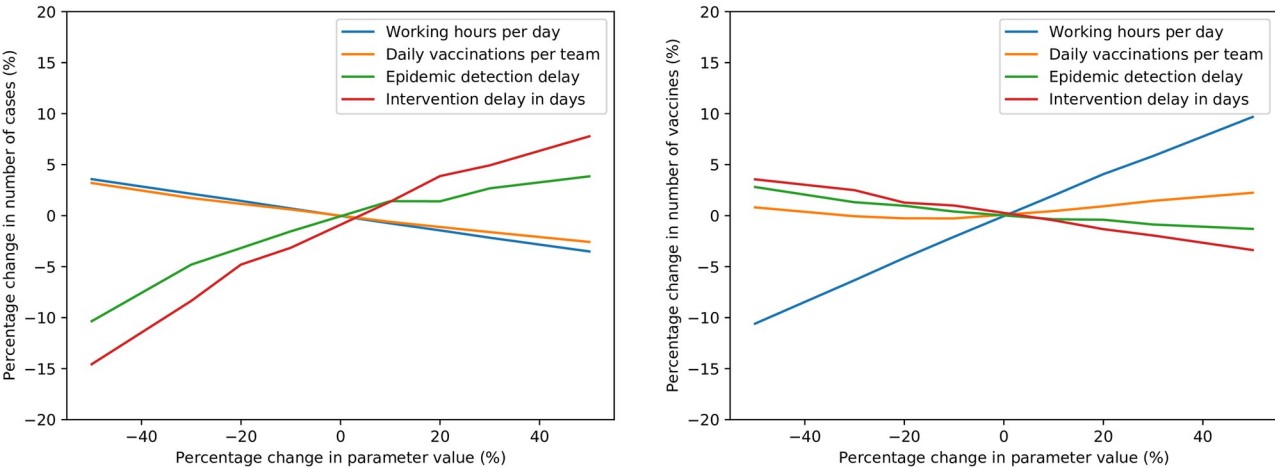

**Fig 7. Sensitivity of average cases (left) and vaccinations (right) to changes in intervention parameters.**

1.5%, for the range considered. For the polycentric, city, and rural networks modelled, this daily increase in total cases is 0.4%, 0.6%, and 2.7%, respectively. These values were obtained as the respective slopes of least-squares linear regression approximations of each network's sensitivity plot for intervention delay, as depicted in Fig 8. It is clear that reducing the intervention delay, by even a single day, can make a large difference in the number of total cases and deaths in an epidemic. As this difference is most significant for rural networks, the use of drones to perform prompt deliveries to hard-to-reach locations is an even more attractive prospect.

The effects of changes in the number of response teams are depicted separately to the remainder of the parameter set, in Fig 9—since the results merit further discussion, and are presented for both an untargeted and a targeted intervention. As expected, as the number of response teams increases, there are fewer cases—with a decreasing marginal benefit as more teams are added. This reduction in marginal benefit, which is most pronounced for targeted interventions, indicates that the benefit of using an additional team in an intervention should be weighed against the marginal cost thereof. The effect of changes in the number of teams on vaccinations is, however, less intuitive. As expected, the addition of more teams results in slightly more vaccinations being possible, for both vaccination types. However, as the number of teams reduces, there is a sharp increase in the number of vaccines required for the untargeted intervention, while the targeted intervention's vaccinations continue to decrease. A likely hypothesis for this is that the reduction in teams, and ineffective untargeted vaccination, results in a failure to timeously control the epidemic. This perhaps-surprising result indicates that a failure to effectively control an epidemic initially may result in an increase in cost required to eventually do so. Furthermore, as previously discussed, this highlights the superiority of targeted vaccination to untargeted vaccination.

**Network parameters.** The sensitivities of the result metrics to changes in three network-related parameters are depicted in Fig 10. As the amount of migration (i.e. the migration multiplier $k_m$) is increased, there is a small increase in the average number of cases, and a small decrease in the number of vaccinations. This result is expected, since increased migration spreads the epidemic more quickly. Further investigation into the impact of reduced migration, perhaps using a more accurate migration model or agent-based simulation, may be insightful. As the diameter of the monocentric network increases, there is a slight decrease in the number of cases (since there is less migration between locations), and an even slighter

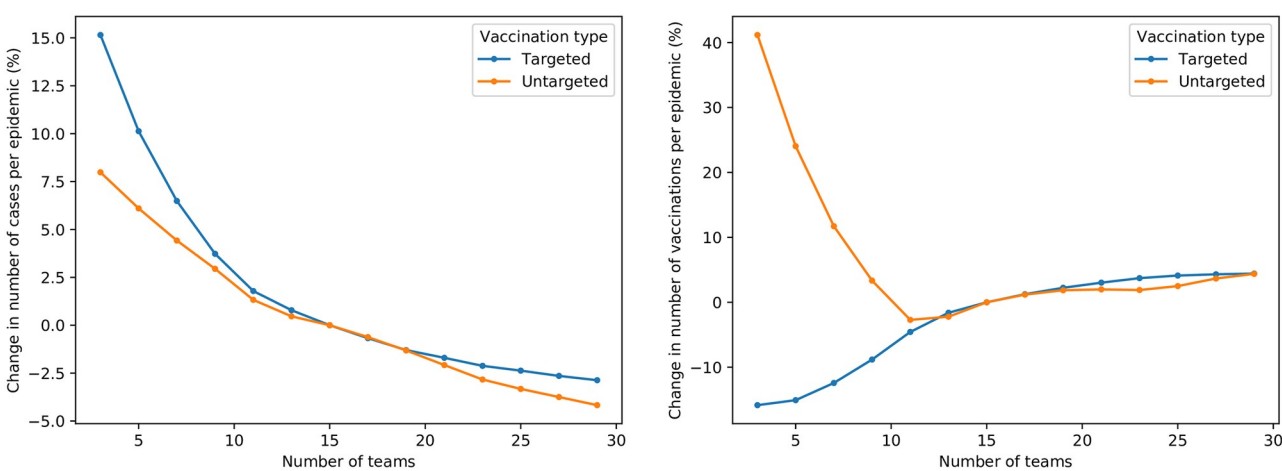

**Fig 8. Effect of intervention delay on cases, for various network structures.**

**Fig 9. Sensitivity of average cases (left) and vaccinations (right) to changes in number of response teams.**

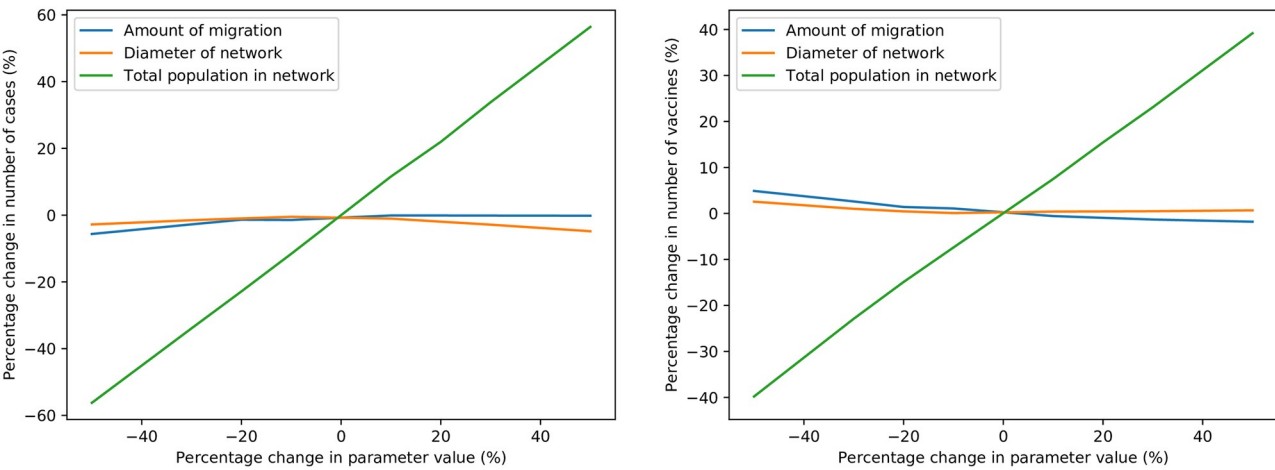

**Fig 10. Sensitivity of average cases (left) and vaccinations (right) to changes in network parameters.**

reduction in the number of vaccinations. The larger the network, the longer drone deliveries take; which results in marginally lower vaccine stock levels. Naturally, increases in the total population of the network cause a proportional increase in the number of cases and vaccinations—although, the former increases comparatively more than the latter.

An additional network-related parameter, the population vaccination rate at the start of an epidemic, obviously plays a significant role in the progression thereof. In simulations, it was found that, for each percentage point increase in the vaccination rate within the range considered, the number of cases reduces by 3.32%, and the number of vaccinations by 0.27%. That is, if the epidemic even occurs at all—the higher the vaccination rate, the less likely an epidemic, and its resultant cost, becomes. Consequently, it is imperative that routine vaccination activities achieve higher population vaccination rates.

**Drone parameters.** A range of drone-related parameters were tested, to determine the sensitivity of results to changes in their values. The results found were fairly intuitive; as either the drone capacity or average speed increases, the number of possible vaccinations increases—and subsequently, the number of cases decreases. Conversely, as the launch time per flight increases, the number of vaccinations reduces, and the number of cases increases. Although these results are expected, the extent to which these seemingly unimportant parameters affect the number of cases is notable. This is the case because, with a limited number of drones, every hour of delivery time is valuable. A 50% decrease of the flight launch time (from 10 minutes, to 5 minutes) results in more than a 5% increase in the number of vaccinations, and a reduction in total cases by over 2%. If the average speed per flight is reduced from 100km/h to 50km/h, the mean total number of cases per simulation increases by 2.4%. Similarly, if the number of vaccines delivered per flight is reduced from 60 to 30, the number of cases increases by 3.5%. These results indicate that, due to their increased speed, fixed-wing drones are more effective than quadcopter drones for prompt vaccine delivery—particularly for larger networks. It seems also that a sense of urgency must be maintained when performing deliveries, as every minute wasted can have an impact on the number of cases. These results do not, however, take into account response teams' potential need for travel time between locations, particularly in rural networks with hard-to-reach populations where team availability may be a constraint, instead of vaccine stock.

An additional parameter relating to drones was also investigated; the number of drones available for deliveries. This is a parameter of such importance in an intervention using drone

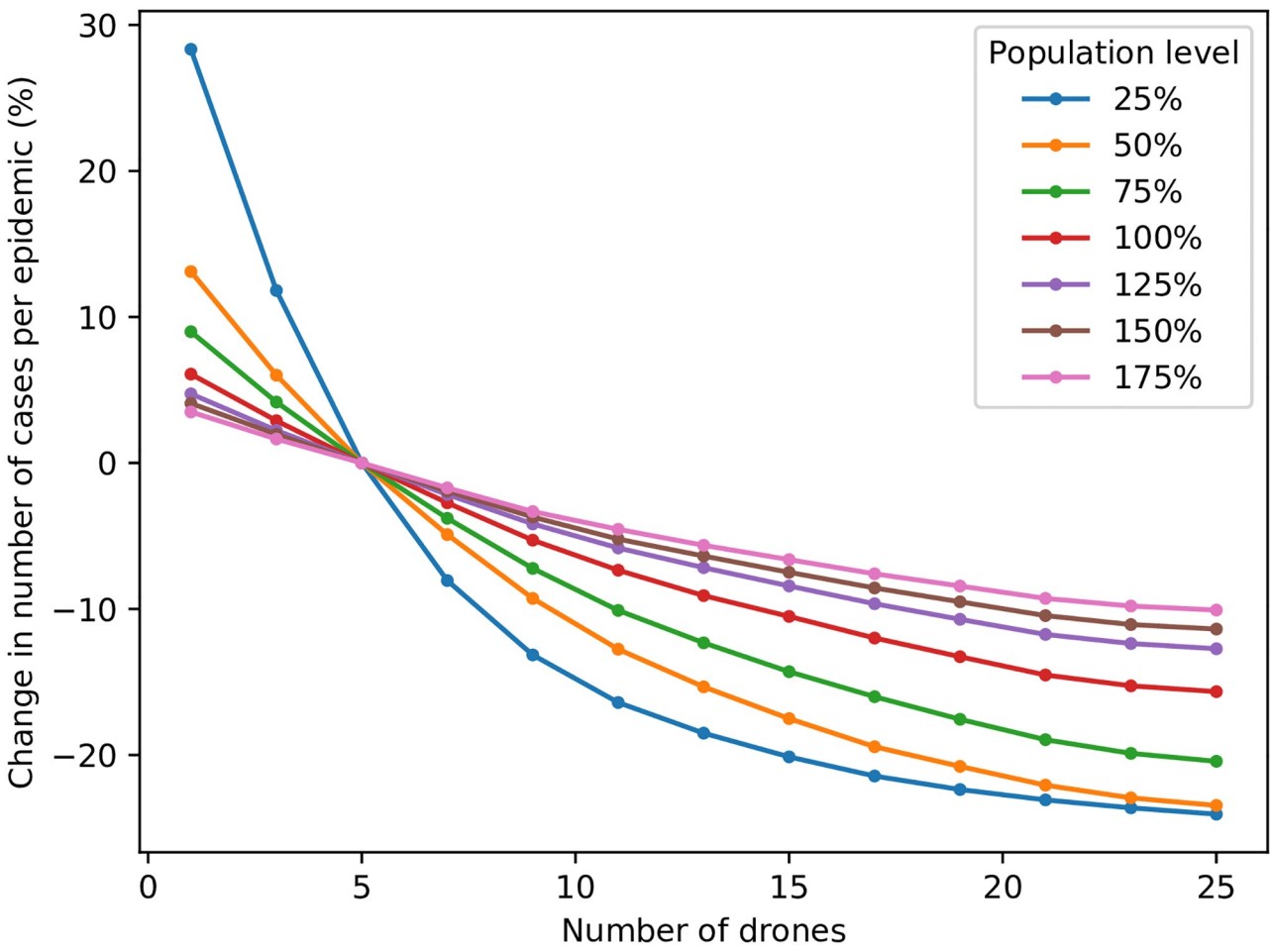

**Fig 11. Impact of number of drones on cases, for various population levels.**

delivery that results are presented separately, for a variety of population levels in the mono-centric network, in Fig 11. This allows the impact of increased population size on the number of required drones to be seen clearly. As expected, as the number of drones is increased from the default value of 5, the number of cases decreases—albeit with a much-reduced marginal benefit as more and more drones are added. It is also clear that the benefit of adding more drones is greater for networks with a smaller population, which is expected; the addition of a drone for a network with triple the population naturally affects proportionately fewer members of that network's population. The number of cases for most population levels appears to reach an equilibrium of sorts when 15 or more drones are used, although providing a general recommendation would be misleading, since the number of drones required is strongly dependent upon the population level and diameter of a network.

## Conclusion

This article introduced some of the logistical challenges faced in epidemic interventions, and the possibility of using drones for the delivery of vaccines and other medical supplies, in the context of an epidemic. A network-based SEIRVD epidemic model was formulated, to investigate the feasibility of drone delivery, and the effect of certain resource allocation strategies and parameters of interest on an epidemic. Repeated simulations were performed using this

model, with four generic network structures as input; representing common population distributions. This allowed the results to be more general than if specific case studies had been considered.

Two resource allocation models were formulated and integrated with this simulation model. These models maximise expected prevented exposures (EPE), by optimising daily resource allocation decisions during each simulation. In order to evaluate the effectiveness of these models, they were compared with a variety of alternative resource allocation strategies, on the basis of cases and vaccines used.

The simulation model was validated on a 2003-04 measles epidemic in Niger, and proved to replicate the actual epidemic's propagation accurately. Initial simulations with each of the four network structures indicated that using targeted vaccination reduces cases significantly, and requires approximately 50% fewer vaccinations to do so—a result which highlights the importance of documenting the vaccination history of individuals well.

Both EPE resource allocation models formulated in this article were shown to be highly effective in reducing epidemic cases, as well as vaccine usage—particularly for targeted interventions. This is the case for all network types, although to varying extents; the efficiencies of the strategies considered appear somewhat inconsistent across network types. In the comparison of team allocation strategies, under the assumption of unlimited vaccine availability, the EPE team strategy performed above-average for all networks, for both targeted and untargeted interventions. In fact, for the former, the EPE strategy resulted in the minimum number of cases for all network types. Besides this, team allocation proportionally to total population and susceptible population were shown to be fairly effective strategies. Conversely, allocating teams uniformly, or proportionally to the number of infections per location, is ineffective, and results in an increased number of cases and vaccines required. When the vaccine delivery strategies were compared by aggregating results over the team strategies, the EPE delivery strategy was again found to be superior to the alternatives. With the exception of the rural network with untargeted vaccination, the EPE delivery strategy resulted in the minimum number of cases and vaccinations, for all networks and for both targeted and untargeted vaccination. Both delivery strategies prioritising locations according to susceptible population performed consistently well, whereas, in accordance with existing literature, the strategies prioritising total population and infections performed poorly, usually resulting in higher costs and more cases. It is important to note, though, that since the effectiveness of the strategies varies according to network type, it is important to consider a network's population distribution when making resource allocation decisions.

Simulations performed for sensitivity analysis revealed the devastating impact which each day of delay before epidemic intervention causes. Increases in both epidemic detection delay, and intervention delay, result in a notable increase in epidemic cases. For the latter, a single further day of delay was shown to increase cases by between 0.4% and 2.7%, depending on the network structure. The urgency of epidemic intervention is already well known, and this result confirms that notion. The impact of a failure to promptly control an epidemic, with insufficient response teams, was also demonstrated; with fewer teams, there are inevitably more cases, and an increase in cost when vaccination is untargeted. In addition to these conclusions, the importance of routine vaccination activities was highlighted by the result that, for each percentage point increase in the pre-epidemic vaccination rate within the range tested, there's a 3.32% decrease in the average number of epidemic cases.

Sensitivity analysis on four drone-related parameters revealed that, due to their superior speed and range, fixed-wing drones are more appropriate for epidemic response than quadcopters, for the network types considered; a reduction of the average drone speed from 100km/h to 50km/h caused a 2.4% increase in the average simulated number of cases. It is

difficult to provide a general recommendation for how many drones should be used in an epidemic response, since this depends on the population, network type and drone capacity. For the fixed-wing drones and networks considered though, 15 drones appear to suffice for most population levels; although the more, the better.

Potential avenues for future work and extension upon this article include: the use of agent-based simulation to model migration and interpersonal disease propagation more accurately; forecasting EPE to an arbitrary future date instead of one day ahead; random, parameterized generation of network structures, instead of only considering four specific ones; and constraining vaccine availability at the DC.

To reiterate the conclusions drawn, and provide a recommendation for improved epidemic response thereon; the EPE models presented were the most effective resource allocation strategies for reducing cases, as well as cost. They are thus recommended for implementation in epidemic interventions. In the absence of such models, resource allocation according to susceptible population, rather than infections or total population, is best. Further, since targeted vaccination is superior to untargeted vaccination, and each day of intervention delay notably affects an epidemic's spread, a targeted intervention with minimal delay is advisable. Finally, fixed-wing drones are a viable option for vaccine delivery in an epidemic intervention, if a sufficient number of them are used—and may be tremendously helpful in areas with poor road infrastructure.

## Supporting information

**S1 Fig. Resource allocation strategy comparisons for the polycentric network structure.** Results for the untargeted intervention are depicted on the left, and results for the targeted intervention are on the right.
(TIF)

**S2 Fig. Resource allocation strategy comparisons for the city network structure.** Results for the untargeted intervention are depicted on the left, and results for the targeted intervention are on the right.
(TIF)

**S3 Fig. Resource allocation strategy comparisons for the rural network structure.** Results for the untargeted intervention are depicted on the left, and results for the targeted intervention are on the right.
(TIF)

**S4 Fig. Generic network structure maps.** The population size at each network location is represented by the radius of the dot, and the epicentre is marked in red. The monocentric network is top left, the polycentric network is top right, the city-type network is bottom left, and the rural-type network is bottom right.
(TIF)

**S5 Fig. Base simulation progression plots.** The below plots depict the progression of a single simulated measles epidemic, in each of the four network types considered. Each of the four simulations were performed with the default set of parameters, and untargeted vaccination. The plots each give an indication of how the total network population progresses between the S, E, I, R, V, and D categories in that network. The monocentric network is top left, the polycentric network is top right, the city-type network is bottom left, and the rural-type network is bottom right.
(TIF)

**S1 Table. Input dataset for monocentric network structure.**
(PDF)

**S2 Table. Input dataset for polycentric network structure.**
(PDF)

**S3 Table. Input dataset for city network structure.**
(PDF)

**S4 Table. Input dataset for rural network structure.**
(PDF)

**S5 Table. Input dataset for Niamey epidemic validation.**
(PDF)

## Acknowledgments

We primarily acknowledge God, for without Him any lives saved in epidemic interventions are purposeless and ephemeral. Therefore, above all, we thank Him for the eternal life we are offered through Jesus. We thank James Mba Azam and Prof. Juliet Pulliam for introducing us to the idea of vaccine delivery using drones, and for the useful consultations and literature references regarding the challenges faced in the vaccine supply chain. We thank our families for their continual support, and Matter specifically thanks Daniella Neville and Brandon Coetzer, for the inspiration to strive for excellence.

## Author Contributions

**Conceptualization:** Dean Matter, Linke Potgieter.

**Data curation:** Dean Matter.

**Formal analysis:** Dean Matter.

**Funding acquisition:** Linke Potgieter.

**Methodology:** Dean Matter, Linke Potgieter.

**Project administration:** Dean Matter.

**Software:** Dean Matter.

**Supervision:** Linke Potgieter.

**Validation:** Dean Matter, Linke Potgieter.

**Visualization:** Dean Matter.

**Writing – original draft:** Dean Matter.

**Writing – review & editing:** Dean Matter, Linke Potgieter.

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
