## [Decision Letter · Decision Letter 0]

24 Dec 2020

PONE-D-20-28939

Allocating epidemic response teams and vaccine deliveries by drone in generic network structures, according to expected prevented exposures

PLOS ONE

Dear Dr. Matter,

Thank you for submitting your manuscript to PLOS ONE. After careful consideration, we feel that it has merit but does not fully meet PLOS ONE’s publication criteria as it currently stands. Therefore, we invite you to submit a revised version of the manuscript that addresses the points raised during the review process.

We look forward to receiving your revised manuscript.

Kind regards,

Gabriele Oliva, Ph.D

Academic Editor

PLOS ONE

Journal Requirements:

2. Please upload a copy of Supporting Information Tables S5, S6 and S7 which you refer to in your text on page 23.

Additional Editor Comments:

Two reviews were collected, both suggesting minor revision. After carefully reviewing the manuscript myself, I confirm this judgement.

Reviewers' comments:

Reviewer's Responses to Questions

**Comments to the Author**

1. Is the manuscript technically sound, and do the data support the conclusions?

Reviewer #1: Yes

Reviewer #2: Yes

2. Has the statistical analysis been performed appropriately and rigorously? 

Reviewer #1: Yes

Reviewer #2: N/A

3. Have the authors made all data underlying the findings in their manuscript fully available?

Reviewer #1: Yes

Reviewer #2: Yes

4. Is the manuscript presented in an intelligible fashion and written in standard English?

Reviewer #1: Yes

Reviewer #2: Yes

5. Review Comments to the Author

Reviewer #1: The authors formulate a network-based SEIRVD epidemic model and propose response team and vaccine delivery allocations according to expected prevented exposures. They compare their models to alternative resource allocation strategies and show their proposed strategies reduce the number of cases and number of vaccines required. Overall, the analysis looks plausible and well executed, however, I have a few comments and suggestions.

1.) Given that the majority of the results presented compare untargeted and targeted vaccination, the reader may benefit from a more detailed explanation of how the vaccinated population for recovered individuals is incorporated in the SEIRVD equations (Eq 1-6) under both the targeted and untargeted vaccination strategies.

2.) The parameter delta_i first appears in the text on page 8 in constraint (11) of the delivery allocation model, yet the formal definition for this parameter is not given until page 11, right before the Results section. I suggest either moving this formal definition to be included in the model description or provide a brief description of what delta_i defines in the model description.

3.) Are the parameter values defined in Table 1 specific to the measles pandemic? If so, I suggest mentioning this in the Parameters section and how the values would need to be adapted for other pandemics.

4.) Tables S4, S5, and S9 define a location CBD but this acronym is never clearly defined. Please clarify.

5.) It is not clear from the S10 Fig caption whether the results are from targeted or untargeted vaccination. It appears as if they are untargeted (based on Table 2) but the numbers do not seem to align between Table 2 and S10 Fig (i.e., vaccinations for the city network type is 1.2 million in S10 Fig but 1.25 million in Table 2). Please clarify this discrepancy. If S10 Fig is just from a single simulation instance, then this should be stated in the figure caption.

6.) Please include which plot corresponds to untargeted and targeted vaccination in the captions for S1, S2, S3 Figures.

7.) An acknowledgement and discussion of any limitations in your research should be included in the Conclusion section.

Reviewer #2: The paper introduced a method for addressing logistical challenges in an epidemic intervention context, such as vaccine shortages and poor road infrastructure. Specifically, the authors explored the feasibility of using drones to deliver vaccines and other medical supplies in order to develop a vaccine allocation strategy coupled with a team allocation strategy. These models maximize expected prevented exposures (EPE), optimizing day-to-day resource allocation decisions. Multiple network structures are evaluated with the resource allocation models to represent interconnected populations over which simulations are performed. The paper is overall interesting and well structured, but I would report below some comments that may help the authors to strengthen their work:

- The analysis of vaccine delivery allocation is based on the constraint of limited daily delivery time, but there is no mention of limits regarding the maximum distances that drones can travel.

- It is specified that vaccines deliveries to locations are dependent on the number of teams allocated there, but in eq. 10-11-12-13 there is no coupling term with the problem developed in the previous section, regarding the strategy for team allocation.

- It might be interesting to remove in the calculation of EPE the assumption that the number of vaccines in stock is unlimited since on page 13 it is pointed out that generally vaccines are very expensive and in short supply.

- At the beginnig, the paper appears to be set in the context of the current Covid-19 outbreak, but later only discusses data from the measles outbreak. Is it therefore possible to adapt the same analysis to different epidemics, or is it necessary to modify the models?

- Equation 11 related to the constraint on the times turns out in general unclear. Please edit for clarity explaining the terms.

- It is not very clear whether the untargeted or targeted vaccination strategy is chosen for the analysis and why.

6. PLOS authors have the option to publish the peer review history of their article (what does this mean?). If published, this will include your full peer review and any attached files.

Reviewer #1: No

Reviewer #2: No

---

## [Author Response · Author response to Decision Letter 0]

2 Feb 2021

The paper has been revised according to the reviewers’ remarks. The authors would like to sincerely thank the reviewers for their comments, and the time taken to review the paper. Please find below the list of reviewers’ comments, each accompanied by a response from the authors.

Reviewer 1:

The authors formulate a network-based SEIRVD epidemic model and propose response team and vaccine delivery allocations according to expected prevented exposures. They compare their models to alternative resource allocation strategies and show their proposed strategies reduce the number of cases and number of vaccines required. Overall, the analysis looks plausible and well executed, however, I have a few comments and suggestions.

1.) Given that the majority of the results presented compare untargeted and targeted vaccination, the reader may benefit from a more detailed explanation of how the vaccinated population for recovered individuals is incorporated in the SEIRVD equations (Eq 1-6) under both the targeted and untargeted vaccination strategies.

Response: Thank you for this valuable observation. We have amended our description of the treatment of recovered individuals under targeted and untargeted interventions, respectively – and realized we made a significant omission from the SEIRVD equations. We had accidentally omitted the number of recovered individuals vaccinated on day t at each location i, which is now represented by the quantity vi,tθRi,t. This quantity is added to Vi,t+1 and subtracted from Ri,t+1, in equations (5) and (4), respectively. Fortunately, this omission was only from the described formulation of the equations and not from the implementation thereof – and as a result, all results remain valid and unchanged. We have added a description of this new quantity in the final paragraph of the subsection titled “Epidemic model”.

2.) The parameter delta_i first appears in the text on page 8 in constraint (11) of the delivery allocation model, yet the formal definition for this parameter is not given until page 11, right before the Results section. I suggest either moving this formal definition to be included in the model description or provide a brief description of what delta_i defines in the model description.

Response: As suggested, we have now defined delta_i immediately prior to its appearance in the formulation in (10)-(13). This is consistent with the pattern used for other parameters such as c, which is defined prior to (10)-(13), and is only attributed a particular value in the Parameters section.

3.) Are the parameter values defined in Table 1 specific to the measles pandemic? If so, I suggest mentioning this in the Parameters section and how the values would need to be adapted for other pandemics.

Response: The values are indeed measles-specific. We have added a discussion of the parameter values and generality of the SEIRVD model to the Parameters section, and have explicitly stated that the values are specifically set to simulate a measles epidemic. Despite this, the sensitivity analysis performed indicates that the results are not particularly sensitive to changes in these parameters, and as a result, the conclusions are most likely valid for similar epidemics.

4.) Tables S4, S5, and S9 define a location CBD but this acronym is never clearly defined. Please clarify.

Response: We have replaced the occurrences of the acronym “CBD” (Central Business District), with the more appropriate descriptor, “Centre”.

5.) It is not clear from the S10 Fig caption whether the results are from targeted or untargeted vaccination. It appears as if they are untargeted (based on Table 2) but the numbers do not seem to align between Table 2 and S10 Fig (i.e., vaccinations for the city network type is 1.2 million in S10 Fig but 1.25 million in Table 2). Please clarify this discrepancy. If S10 Fig is just from a single simulation instance, then this should be stated in the figure caption.

Response: We have clarified the distinction in the paragraph prior to Table 2, and in the caption of S10 Fig. S10 Fig does only depict the progression curves of a single simulation instance per network, whereas Table 2 contains results aggregated over 100 simulations. The adjustment to the paragraph is quoted below, for the convenience of the reader:

“The aggregated results of these simulations for each network are contained in Table 2, and the figures in S10 Fig depict the example epidemic progression curves for each, for a single simulation in each network.”

6.) Please include which plot corresponds to untargeted and targeted vaccination in the captions for S1, S2, S3 Figures.

Response: We have added captions to S1, S2, and S3 Figures to indicate that results for untargeted vaccination are plotted on the left, and targeted vaccination on the right.

7.) An acknowledgement and discussion of any limitations in your research should be included in the Conclusion section. 

Response: We have added a list of avenues for future work in the penultimate paragraph of the Conclusion section. This list reveals the limitations in the research that we are aware of and propose future investigation into. The list is quoted below: 

“Potential avenues for future work and extension upon this article include: the use of agent-based simulation to model migration and interpersonal disease propagation more accurately; forecasting EPE to an arbitrary future date instead of one day ahead; random, parameterized generation of network structures, instead of only considering four specific ones; and constraining vaccine availability at the DC.”

Reviewer 2

The paper introduced a method for addressing logistical challenges in an epidemic intervention context, such as vaccine shortages and poor road infrastructure. Specifically, the authors explored the feasibility of using drones to deliver vaccines and other medical supplies in order to develop a vaccine allocation strategy coupled with a team allocation strategy. These models maximize expected prevented exposures (EPE), optimizing day-to-day resource allocation decisions. Multiple network structures are evaluated with the resource allocation models to represent interconnected populations over which simulations are performed. The paper is overall interesting and well structured, but I would report below some comments that may help the authors to strengthen their work:

1. The analysis of vaccine delivery allocation is based on the constraint of limited daily delivery time, but there is no mention of limits regarding the maximum distances that drones can travel.

Response: We have specified the drone delivery radius more explicitly in the Parameters section, and have added it to Table 1. Under our simplifying assumption stated in the second paragraph of the Methodology section – that there is only one DC – the diameter of the entire network must be selected to be small enough for drones to be able to fly from the DC to any location and back. The fixed-wing drones under consideration are only able to make single deliveries per flight, and therefore drone deliveries are simply there-and-back, instead of a more complex routing problem. As a result of both of these points, since drone flight range is already taken into account when selecting the DC and all locations in each network can be reached, we do not consider flight range to be a constraint in the vaccine delivery allocation problem as well.

2. It is specified that vaccines deliveries to locations are dependent on the number of teams allocated there, but in eq. 10-11-12-13 there is no coupling term with the problem developed in the previous section, regarding the strategy for team allocation.

Response: In the second paragraph of the subsection, “EPE strategy for vaccine delivery allocation”, it is stated that “[v]accine deliveries to locations are dependent on the number of teams allocated there”. The parameter representing the number of teams at location i on day t is eta_{i,t}, which is the same parameter used in the team allocation problem. We have added this parameter to the aforementioned statement for clarity. To explain the link between the models further; eta_{i,t} forms part of the equation for parameter v`_{i,t}, which in turn forms part of the upper bound (Eq 12) of vaccines delivered per location. In addition to this, eta_{i,t} forms part of the equation for the quantity v_{xi,i,t}, which implicitly affects the EPE value.

3. It might be interesting to remove in the calculation of EPE the assumption that the number of vaccines in stock is unlimited since on page 13 it is pointed out that generally vaccines are very expensive and in short supply.

Response: Considering limited vaccine availability is indeed a valuable area for investigation. We have included a related item in the newly-added list of proposed areas for further research (in the Conclusion section): “constraining vaccine availability at the DC”. While that does not necessarily cover the proposed adjustment to the EPE calculation, it is important to note that this assumption of unlimited availability is only used for the calculation of team allocation EPE, and not for delivery allocation EPE. In the latter, existing vaccine stock is represented by the parameter nu_{i,t}, and forms a significant part of the calculation. Firstly, this assumption is merely present in order to prevent a complex circular dependency between the two problems; if teams are dependent on vaccine availability, and vaccine availability is dependent on teams, the optimization of either becomes difficult. Secondly, on a practical note, vaccines can not necessarily be stored at a point of dispensing unless a team is present, and therefore allocating teams first is necessary. Additionally, it is challenging to quantify an estimated constraint for vaccines in stock at each location without having any teams present to base that constraint upon.

4. At the beginnig, the paper appears to be set in the context of the current Covid-19 outbreak, but later only discusses data from the measles outbreak. Is it therefore possible to adapt the same analysis to different epidemics, or is it necessary to modify the models?

Response: We have amended the first paragraph in the Parameters section to discuss the generality of the SEIRVD model and its ability to simulate other communicable diseases sharing similar properties. As discussed there, it is possible to adapt the analysis to other epidemics by adjusting the parameter values in the model. However, despite the occurrence of a latent period for the virus, attempting to apply the model to COVID-19 may be unhelpful, in the absence of accurate parameter estimates and substantive evidence of permanent post-infection immunity (a property we’ve listed there as a requirement for application of the model).

5. Equation 11 related to the constraint on the times turns out in general unclear. Please edit for clarity explaining the terms.

Response: As requested by the other reviewer also, we have clarified the explanation of Eq 11 by adding a description of the constraint and the parameters therein, prior to the constraint. As described there, the constraint is a limit on the number of deliveries which can be performed to each location, based on the time it takes to make each delivery.

6. It is not very clear whether the untargeted or targeted vaccination strategy is chosen for the analysis and why.

Response: We have clarified the explanation of untargeted and targeted vaccination in the second paragraph of page 5. We state there that, “[untargeted vaccination] is used by default in this article, except where otherwise stated.” The reason for this is cited prior; “[untargeted vaccination is] sometimes used in outbreak response where there is a high risk of the epidemic spreading”. Throughout the results, where it is not indicated or there is no comparison between targeted and untargeted vaccination, untargeted vaccination is employed. We have repeated this assertion – that untargeted vaccination is used by default – in the first paragraph of the Parameters section.

---

## [Decision Letter · Decision Letter 1]

19 Feb 2021

Allocating epidemic response teams and vaccine deliveries by drone in generic network structures, according to expected prevented exposures

PONE-D-20-28939R1

Dear Dr. Matter,

We’re pleased to inform you that your manuscript has been judged scientifically suitable for publication and will be formally accepted for publication once it meets all outstanding technical requirements.

Kind regards,

Gabriele Oliva, Ph.D

Academic Editor

PLOS ONE

Additional Editor Comments (optional):

Both reviewers agree that the revision has successfully addressed their comments. I concur with this evaluation.

Reviewers' comments:

Reviewer's Responses to Questions

**Comments to the Author**

1. If the authors have adequately addressed your comments raised in a previous round of review and you feel that this manuscript is now acceptable for publication, you may indicate that here to bypass the “Comments to the Author” section, enter your conflict of interest statement in the “Confidential to Editor” section, and submit your "Accept" recommendation.

Reviewer #1: All comments have been addressed

Reviewer #2: All comments have been addressed

2. Is the manuscript technically sound, and do the data support the conclusions?

Reviewer #1: (No Response)

Reviewer #2: Yes

3. Has the statistical analysis been performed appropriately and rigorously? 

Reviewer #1: (No Response)

Reviewer #2: N/A

4. Have the authors made all data underlying the findings in their manuscript fully available?

Reviewer #1: (No Response)

Reviewer #2: Yes

5. Is the manuscript presented in an intelligible fashion and written in standard English?

Reviewer #1: (No Response)

Reviewer #2: Yes

6. Review Comments to the Author

Reviewer #1: (No Response)

Reviewer #2: (No Response)

7. PLOS authors have the option to publish the peer review history of their article (what does this mean?). If published, this will include your full peer review and any attached files.

Reviewer #1: No

Reviewer #2: No

---

## [Editor Report · Acceptance letter]

23 Feb 2021

PONE-D-20-28939R1 

Allocating epidemic response teams and vaccine deliveries by drone in generic network structures, according to expected prevented exposures 

Dear Dr. Matter:

I'm pleased to inform you that your manuscript has been deemed suitable for publication in PLOS ONE. Congratulations! Your manuscript is now with our production department. 

Kind regards, 

on behalf of

Dr. Gabriele Oliva 

Academic Editor

PLOS ONE